# Diverse Roles of Protein Palmitoylation in Cancer Progression, Immunity, Stemness, and Beyond

**DOI:** 10.3390/cells12182209

**Published:** 2023-09-05

**Authors:** Mingli Li, Leisi Zhang, Chun-Wei Chen

**Affiliations:** 1Department of Systems Biology, Beckman Research Institute, City of Hope, Duarte, CA 91010, USA; leiszhang@coh.org; 2City of Hope Comprehensive Cancer Center, Duarte, CA 91010, USA

**Keywords:** palmitoylation, cancer, cancer treatment, protein post-translational modification, tumor, ZDHHCs

## Abstract

Protein S-palmitoylation, a type of post-translational modification, refers to the reversible process of attachment of a fatty acyl chain—a 16-carbon palmitate acid—to the specific cysteine residues on target proteins. By adding the lipid chain to proteins, it increases the hydrophobicity of proteins and modulates protein stability, interaction with effector proteins, subcellular localization, and membrane trafficking. Palmitoylation is catalyzed by a group of zinc finger DHHC-containing proteins (ZDHHCs), whereas depalmitoylation is catalyzed by a family of acyl-protein thioesterases. Increasing numbers of oncoproteins and tumor suppressors have been identified to be palmitoylated, and palmitoylation is essential for their functions. Understanding how palmitoylation influences the function of individual proteins, the physiological roles of palmitoylation, and how dysregulated palmitoylation leads to pathological consequences are important drivers of current research in this research field. Further, due to the critical roles in modifying functions of oncoproteins and tumor suppressors, targeting palmitoylation has been used as a candidate therapeutic strategy for cancer treatment. Here, based on recent literatures, we discuss the progress of investigating roles of palmitoylation in regulating cancer progression, immune responses against cancer, and cancer stem cell properties.

## 1. Introduction

Discovered in the early 1970s [1,2,3,4], protein S-palmitoylation (hereinafter referred to as protein palmitoylation) is a type of protein post-translational modification. It refers to the reversible covalent attachment of a fatty acyl chain—specifically, a 16-carbon palmitate acid—to the specific cysteine residues on target proteins via a thioester bond (Figure 1). By attaching the saturated fatty acid to proteins, palmitoylation increases the hydrophobicity of proteins and plays critical roles in regulating protein stability, interaction with effector proteins, sub-cellular localization, enzymatic activity, membrane trafficking, and many other aspects of cellular processes. The reversible nature of palmitoylation enables palmitoylation to impact protein function in a spatiotemporal and dynamic manner. For example, it has been reported that palmitoylation enables ionotropic glutamate receptors, responsible for the glutamate-mediated postsynaptic excitation of neurons, to interact with their downstream partners and, thus, transduce signaling by attaching their tails to the plasma membrane [5]. Palmitoylation has also been shown to protect the protective antigen receptor, which is involved in anthrax toxin, against premature degradation by blocking ubiquitination [6].

The dynamic cycle of palmitoylation takes seconds to hours to modulate protein biological functions [7,8,9,10]. Identification of the palmitoyl transferases, the enzymes that govern protein palmitoylation, was accomplished in 2002 [11,12]. Since these enzymes feature a conserved Asp-His-His-Cys (DHHC) motif responsible for their catalytic activity, they are also known as zinc finger DHHC-type containing (ZDHHC) proteins. To date, 23 different ZDHHCs [8,13], designated as ZDHHC1 to ZDHHC24 and skipping ZDHHC10, have been identified in mammalian cells. They are polytopic membrane proteins, and most of them are located in the ER or the Golgi membrane, with ZDHHC5, 20, and 21 positioned in the plasma membrane [14,15]. ZDHHC isoforms consist of four to six transmembrane (TM) domains. The catalytic ZDHHC domain is located in the cytosolic part in between the second and third TMs [16]. The ZDHHCs catalyze protein palmitoylation in a two-step process, which involves auto palmitoylation of themselves to form an acyl–enzyme intermediate and the subsequent transfer of Acyl-CoA to the targeted cysteine residues in substrate proteins [17]. Despite the similar zinc-finger-like motif, ZDHHCs display distinct preferences in substrate proteins and uneven levels of catalytic efficiency [18,19,20]. Palmitoylated proteins may respond to more than one ZDHHC enzyme; however, one particular ZDHHC enzyme often has a stronger effect than others on substrate palmitoylation in the cell. The regulatory mechanisms underlying how ZHHDCs select candidates for modification and their functional redundancy are not entirely clear.

On the contrary, the enzymatic thioester hydrolysis that removes palmitate from palmitoylated proteins is catalyzed by a family of serine hydrolases, including acyl-protein thioesterases (APT1, APT2) [21,22,23], palmitoyl protein thioesterases (PPT1, PPT2) [24,25], α/β hydrolase domain-containing proteins 17A/B/C (ABHD17A/B/C) [26,27,28], and ABHD10 [29]. While APT1 and APT2 are responsible for depalmitoylation of the cytosolic proteins [21,23], PPT1 and PPT2 control protein degradation in lysosomes [30]. The binding affinity of APT1 and APT2 on palmitoylated proteins and their catalytic rates rely on the amino acid sequences surrounding the palmitoylation sites [31,32,33]. Further, although significant overlapping of substrates was observed in APT1 and APT2, selectivity in depalmitoylating substrates was also identified. In addition, this selectivity observed in APTs is mainly achieved through hydrolysis rate but not binding affinity. For example, APT1 has a better hydrolysis rate to short peptide than APT2.

Several methodologies have been developed for detecting palmitoylated proteins. Among these, acyl-biotin exchange (ABE) [34] and acyl-resin-assisted capture (acyl-Rac) [35] are widely used (Figure 2). In these methods, free cysteines of palmitoylated proteins are capped and thioester linkages are subsequently cleaved to generate new thiols. These thiols are then selectively labeled by biotin for ABE or thiol-reactive resin for acyl-Rac, allowing further enrichment and detection of palmitoylated proteins. A variant of this method called acyl-PEG exchange (APE) exploits PEG labeling of newly generated thiols as a mass tag for mobility-shift-based assays to identify levels of protein palmitoylation [36]. Other methods developed for analysis of specific palmitoylated proteins include in-cell imaging based on bio-orthogonal fatty acid labeling and in situ proximity ligation or quantification of palmitoylation levels by gas/liquid chromatography and mass spectrometry [37,38].

A broad spectrum of proteins including enzymes, cancer promoters, cancer suppressors, and transcription factors have been identified to be palmitoylated. Palmitoylome-proteomic studies have identified thousands of palmitoylated proteins [10,39]. The ‘SwissPalm’ database indicates that more than 10% of the entire human proteome is susceptible to palmitoylation [40,41]; among those, over 1000 substrates have already been characterized experimentally. More recently, it has also been shown that among 299 cancer drivers identified in humans, 79 are palmitoylated [8,42]. Functional investigations have shown that palmitoylation is essential for the pathophysiological functions of the many oncoproteins and tumor suppressors, such as RAS, EGFR, and Hedgehog [43,44,45]. Consequently, dysregulation of protein palmitoylation has been implicated in all aspects of cancer hallmark functions, cancer metabolism, and regulation of tumor micro-environment [46,47,48].

Due to the functions of ZDHHCs in palmitoylating cancer-related proteins, roles of palmitoyl transferases in cancer have also been revealed. Aberrant ZDHHC activity has also been shown to be strongly correlated with various types of cancer [49]. For example, copy number variation of a chromosome region including the gene of ZDHHC11 is observed in lung and bladder cancers [50]; upregulation of ZDHHC9 has been found in colorectal tumors [51]; and upregulation of ZDHHC5, as an oncogenic factor, was reported in p53-mutant gliomas [52]. In breast cancer, patients associated with an elevated expression of ZDHHC3 were correlated with lower patient survival [53]. Further, functional investigations have also validated the essential roles of ZDHHCs in cancer. For example, an RNAi-based screen targeting, all 23 members of the ZDHHC family in non-small lung cancer (NSCLC) identified ZDHHC5 to be essential for the growth of NSCLC but not normal human bronchial epithelial cells [54]. In gliomas, inactivation of ZDHHC15 blocked glioma cells proliferation by decreasing activation of Signal Transducer and Activator of Transcription 3 (STAT3) [55] and knockdown of ZDHHC12 reduced the growth, migration, and invasion capabilities of glioma cells [56,57]. In pancreatic cancer, knockdown of ZDHHC3 impaired tumor progression in a xenograft mice model [58], whereas inactivation of ZDHHC9 modified the tumor microenvironment from an immunosuppressive to a proinflammatory environment and, thus, suppressed tumor growth [59]. DHHC9 is also essential for leukemogenesis by palmitoylating oncogenic NRAS [60].

Study of palmitoylation is often conducted from two aspects: palmitoylated proteins and palmitoyl transferases, the enzymes modulating palmitoylation (Figure 1b,c). When the studies are focused on the palmitoylated proteins, several questions can be addressed: 1—if a specific protein is palmitoylated; 2—where the palmitoylation sites are or which cysteine residues are palmitoylated; 3—which ZDHHC is the main palmitoyl transferase responsible for the palmitoylation of the specific protein; 4—how palmitoylation affects functions of the specific protein; 5—if palmitoylation leads to pathological conditions or contributes to physiological conditions by modulating protein function; 6—how the palmitoylation is regulated. When the studies are focused on ZDHHCs, questions to be addressed are as follows: 1—what the biological function of the ZDHHCs is in physiological or specific pathological contexts; 2—for a particular ZDHHC, what the substrates are; 3—if inhibitors can be developed to target the ZDHHCs; 4—how expression of ZDHHCs and ZDHHCs-mediated palmitoylation are regulated. Despite some reviews summarizing functions of palmitoylated proteins in cancer [8,61,62], a review discussing the recent progress on understanding roles of palmitoylation in cancer immunology and cancer stem cell maintenance is lacking. Here, focusing on literatures that were published in the last five years, we discuss the critical roles of palmitoylation in cancer, speculate the future direction of the palmitoylation research field, and discuss the obstacles the field is facing.

## 2. Palmitoylation in Cancer

### 2.1. Palmitoylation in Regulating Growth Signaling

#### 2.1.1. Regulation of AKT Signaling by Palmitoylation

Activated by phosphatidylinositol-3-kinase (PI3K), AKT plays a central role in a variety of cellular events including growth, proliferation, glucose uptake, metabolism, and cell survival [63]. Not surprisingly, a variety of human cancers exhibit upregulated AKT activity and several mouse models with activated AKT develop cancer. Palmitoylation of AKT at cysteine 344 (Cys344) was identified in both HEK293T and preadipocyte HeLa Kyoto cells, which indicated a new layer of regulation for AKT [64]. Mutation of Cys344 resulted in reduction of AKT308 phosphorylation and recruitment of AKT to lysosomes, a process stimulated by inducers of oxidative stress and autophagy (Figure 3a). These observations indicate that palmitoylation is essential for AKT activation and prevents degradation of AKT. Studies to understand roles of AKT palmitoylation in cancer initiation and progression, and to identify the palmitoyl transferase that catalyzes AKT palmitoylation, would be of interest.

In addition to palmitoylating AKT directly to increase AKT signaling, palmitoylation is able to regulate AKT signaling in an indirect manner. For example, by attenuating the stability of mTOR, a kinase complex that phosphorylates AKT, palmitoylation of mTOR mediated by ZDHHC22 reduced AKT signaling in breast cancer cells [65]. Site-directed mutagenesis identified Cys361 and Cys362 as the main palmitoylation sites of mTOR. On the contrary, in liver cancer, palmitoylation of proprotein convertase subtilisin/kexin type 9 (PCSK9), a key enzyme regulating cholesterol metabolism, increased affinity of PCSK9 in binding with phosphatase and tensin homolog (PTEN) [66]. Consequently, binding of PCSK9 led to degradation of PTEN and, thus, released its inhibitory function on AKT signaling. Together, regulation of AKT is context-dependent and it is mediated in several layers.

#### 2.1.2. Regulation of Wnt Signaling by Palmitoylation

Wnt signaling is essential for organ formation during development and for organ homeostasis in postnatal life [67,68]. In osteosarcoma, the most frequent malignant primary bone tumor in children and young adults, ZDHHC19 was found to be highly expressed, and knockdown of ZDHHC19 led to downregulated expression of Wnt/β-catenin [69] (Figure 3b). These observations established the connection between ZDHHC19 and Wnt/β-catenin signaling.

Downstream of Wnt, low-density lipoprotein receptor-related proteins 5 and 6 (LRP5/6) act as co-receptors to activate β-catenin pathway. Palmitoylation of LRP6 was reported in 2008 [70]. Mutation of Cys1394 and Cys1399, two LRP6 palmitoylation sites, diminished the distribution of LRP6 to the plasma membrane by retaining LRP6 in the endoplasmic reticulum (ER). As a result, due to the lack of LRP6 on the plasma membrane, Wnt/β-catenin signaling cannot be transduced efficiently. These data revealed the second layer of regulation on Wnt/β-catenin signaling provided by palmitoylation.

Wnt signaling can be antagonized by dickkopf1 (DKK1), which binds and, thus, degrades LRP6, one of the Wnt receptors [71,72]. Furthermore, DKK1 and LRP6 can form a ternary complex with cytoskeleton-associated protein 4 (CKAP4), a cell surface receptor that activates PI3K/AKT signaling, to enhance DKK1-dependent cancer cell proliferation [73,74,75]. In human cervical carcinoma cells (HeLa), CKAP4 was identified as a substrate of ZDHHC2 from the palmitoylome scale [76]. Palmitoylation of CKAP4 at Cys100 was confirmed by ABE assay, and the direct interaction between ZDHHC2 and CKAP4 was confirmed by Co-immunoprecipitation (Co-IP) (Figure 3b). Mechanistically, palmitoylation by ZDHHC2 is required for CKAP4 trafficking from the ER to the plasma membrane and, thus, mediates PI3K/AKT signaling [77,78]. Indeed, expression of CKAP4-WT but not CKAP4-C100S rescued the reduction in tumor growth induced by knockout of CKAP4. Interestingly, DKK1 induces depalmitoylation of CKAP4 through APT1/2 [79]. Together, palmitoylation of CKAP4 translocated CKAP4 to the plasma membrane, enabling DKK1 to antagonize Wnt signaling.

#### 2.1.3. Regulation of IGF-1/IGF-1R Signaling by Palmitoylation

Flotillin-1 (FLOT-1) is a lipid raft-associated protein that has been implicated in the progression of cancers [80,81,82] and insulin signaling to trigger glucose transporter redistribution in adipocytes [83]. In the process of understanding how FLOT-1, a non-transmembrane protein, sustains insulin signaling on plasma membrane (PM), Morrow et al. found that plasma membrane association of FLOT-1 requires palmitoylation [84]. Palmitoylation of Flot-1 occurs at Cys-34 in the prohibitin-like domain (PHB) at the N terminus of flotillin (Figure 3c). Further study found that the palmitoylation turnover of FLOT-1 in the plasma membrane was induced by Insulin-like growth factor-1 (IGF-1) [85]. As a result, although the mechanism is not fully characterized, palmitoylation of FLOT-1 mediated by ZDHHC19 prevents desensitization of IGF-1R via endocytosis and lysosomal degradation, leading to excessive IGF-1R-mediated signaling and, thus, increased migration and invasion of cervical cancer cells [86]. Although palmitoylation of IGF-1R was not reported, these studies provided solid evidence showing the importance of palmitoylation in mediating IGF-1/IGF-1R signaling.

### 2.2. Roles of Palmitoylation in Cancer Immunology

#### 2.2.1. Regulation of IFNγ/IFNGR1-Mediated PD-L1/PD-1 Signaling by Palmitoylation

Stimulated by interferon-γ (IFNγ) ligand, IFNγ receptor 1 (IFNGR1) activates JAK/STAT signaling and, thus, induces transcription of programmed death protein ligand 1 (PD-L1) [87] (Figure 4). Downstream of PD-L1, programmed death protein 1 (PD-1), the receptor of PD-L1, delivers inhibitory signals to regulate the balance between T cell activation, tolerance, and immunopathology. PD-1 is expressed on the surface of activated T cells as an inhibitory receptor, while its ligand PD-L1 is mainly expressed in antigen-presenting cells and tumor cells. By binding to its receptor PD-1 on T cells, expression of PD-L1 on tumor cells inhibits T cell activation and, thus, drives the escape of tumor cells from immune surveillance. Therefore, components of the IFNγ/IFNGR1-mediated PD-L1/PD-1 signaling pathway play a critical role in the success or failure of immune checkpoint blockades.

Several studies have reported that palmitoylation provides regulatory roles for IFNγ/IFNGR1 signaling by modifying functions of the components. For example, in colorectal cancer cells, stability of IFNGR1 was found to be negatively regulated by palmitoylation; although, the specific ZDHHC that palmitoylates IFNGR1 was not identified [88]. Mutation of Cys122, the palmitoylation site of IFNGR1, or treatment of a palmitoylation inhibitor 2-bromopalmitate (2-BP) blocked the degradation of IFNGR1. Furthermore, IFNGR1 palmitoylation is also essential for the interaction between IFNGR1 and its binding partner AP3D1. As a consequence, IFNGR1 is able to trigger its downstream immune response signaling to inhibit cancer growth.

Further, palmitoylation modulates the immune response to cancer by directly altering PD-L1 functions. Palmitoylation of endogenous PD-L1 was firstly identified in breast cancer cell lines MDA-MB231 and BT549 [89]. Inhibition of PD-L1 palmitoylation by 2-BP decreased PD-L1’s protein level. Mutation of Cys272 or knockdown of ZDHHC9, the palmitoyl transferase responsible for PD-L1, abolished PD-L1 palmitoylation and decreased PD-L1 cell surface distribution. More importantly, blocking PD-L1 palmitoylation sensitized tumor cells to T-cell killing and, thereby, impaired tumor growth in vivo. These observations indicate that, in breast cancer cells, palmitoylation facilitates immune suppression to cancer by stabilizing and maintaining PD-L1 on the cell surface. Further, palmitoylation of PD-L1 in colorectal cancer and lung adenocarcinoma was also reported [90,91]. Of note, unlike ZDHHC9-mediated PD-L1 palmitoylation in breast cancer cells, palmitoylation of PD-L1 in colorectal cancer cells is mediated by ZDHHC3 but not ZDHHC9, indicating that PD-L1 utilizes multiple ZDHHCs to ensure its palmitoylation in different contexts. Mechanistically, palmitoylation stabilizes PD-L1 by suppressing ubiquitination and degradation in lysosomes, thereby repressing anti-tumor immunity. As a treatment strategy, to increase the immune response against cancer, a designed peptide that contains the palmitoylation region of PD-L1 was developed as a competitive inhibitor to inhibit PD-L1 palmitoylation. More recently, roles of ZDHHC9 in anti-tumor immunity were also revealed in pancreatic cancer [59]. Different from the studies conducted on breast cancer, colorectal cancer, and lung adenocarcinoma, this study found that, in the pancreatic tumor xenograft model, tumors with ZDHHC9 knockdown led to a change in PD-L1 level on the surrounding immune cells. Although it is unclear how ZDHHC9 in the tumor cells affects PD-L1 expression level on the surrounding immune cells, this study reported the non-autonomous function of ZDHHC9 in regulating the cancer immune response for the first time.

In addition to PD-L1, palmitoylation also affects functions of its receptor PD-1. In addition to being expressed on the T cells to function as an inhibitory receptor, expression of PD-1 in cancer cells has also been revealed. Different from the function of PD-1 on T cells, in cancer cells, PD-1 promotes tumor growth independently of the adaptive immune system by modulating mTOR signaling. In 2021, Yao et al. revealed the palmitoylation of PD-1 at Cys192 in a variety of cancer cells [92]. Inhibition of PD-1 palmitoylation by 2-BP blocked the interaction between PD-1 and Rab11, a key molecule in transporting the cargo proteins to the recycling endosomes. As a consequence, decreased storage of PD-1 in recycling endosomes led to increased PD-1 degradation in the lysosomes. These results suggest that palmitoylation attenuates degradation of PD-1 by facilitating transportation of PD-1 to the recycling endosomes. More importantly, palmitoylation of PD-1 is essential for cancer cell growth as it is required for PD-1 to interact with its downstream signaling mediators to transduce signaling. Therefore, palmitoylation of PD-1 in cancer cells promotes cancer cell growth by preventing degradation of PD-1 and promoting its interaction with its downstream signaling mediators. Similar to the competitive inhibitor of PD-L1, a peptide containing the palmitoylation region of PD-1 was developed as a competitive inhibitor for PD-1 palmitoylation. Moving forward, it would be interesting to investigate the possibility of PD-1 palmitoylation in T cells and if palmitoylation affects the function of PD-1 in adaptive immune system.

#### 2.2.2. Palmitoylation Triggers Anti-Immune Response by Sorting Proteins into Extracellular Vesicles

The roles of extracellular vesicles (EV), lipid-enclosed particles that circulate bioactive content, in cancer development and progression have been revealed for decades [93]. Recently, Mariscal et al. demonstrated that, by anchoring proteins to cellular membranes, palmitoylation plays an important role in sorting proteins into the EV [94]. By doing so, the anti-immune response is triggered. For example, by secreting palmitoylated proteins through EV, acute myeloid leukemia (AML) cells activate toll-like receptor 2 (TLR2), a receptor that mediates activation of the immune system [95]. Subsequently, TLR2 induced the differentiation of monocytes into T-cell inhibitory myeloid-derived suppressor cells (MDSC), a type of immunosuppressive cell [96], resulting in a blockage of immune-response-mediated AML cell removal. Indeed, although the identity of the palmitoylated proteins in EV was not identified and it is still unknown how palmitoylation triggers TLR2 activation, AML cells treated with 2-BP lost the capacity to generate MDSC and an immune-suppressive environment for their survival.

#### 2.2.3. Regulation of the Innate Immune Response in Preventing Cancer by Palmitoylation

STING (Stimulator of interferon genes) is an innate immune sensor of immune surveillance of viral/bacterial infection and is involved in the maintenance of an immune-friendly microenvironment to prevent tumorigenesis [97,98,99,100]. Palmitoylation of STING was initially reported in 2016 [101]. Dual mutations of Cys88/91 abolished the palmitoylation of STING and, thus, its role in activating expression of its downstream targets including IRF3, IFNβ, and NF-κB, suggesting that palmitoylation is required for activation of STING. Over-expression of ZDHHC3, ZDHHC7, and ZDHHC15 increased palmitoylation of STING, suggesting that they are responsible for STING palmitoylation. Further, another study found that palmitoylation of STING is essential for STING to interact with the mitochondrial voltage-dependent anion channel VDAC2 and, thus, prevent VDAC2-induced mitochondria dysfunction [102].

### 2.3. Regulation of Cancer Stem Cell Potency by Palmitoylation

Cancer stem cells are responsible for cancer treatment resistance, leading to relapse, disease progression, and eventually systemic disease [103]. Lately, functions of palmitoylation in cancer stem cells have received increasing attention. Mechanistic analysis found that palmitoylation not only affects the self-renewal capacity of cancer stem cells but also affects their tumorigenicity (Figure 5). In the past five years, as the majority of studies were conducted on glioblastoma stem cells (GSCs), we focused on discussing roles of palmitoylation in GSCs here.

Acting downstream of ERK and AKT, glycogen synthase kinase 3β (GSK3β), a serine/threonine protein kinase, is also considered as a key hub for promoting malignancy of GSCs [104,105]. Zhao et al. found that roles of GSK3β are affected by palmitoylation when ZDHHC4 palmitoylates GSK3β at the Cys14 residue [106]. Mechanistically, palmitoylation of GSK3β negatively affected binding of GSK3β to AKT1 and S6K kinases, thereby releasing the inhibitory roles of AKT1 and S6K on GSK3β. Downstream of GSK3β, activated GSK3β phosphorylates histone methyltransferase EZH2, which, in turn, regulates STAT3 methylation and phosphorylation, leading to increased expression of stem-cell-related genes [107]. More importantly, due to the increased GSK3β activity induced by palmitoylation, tumorigenicity of GSCs was increased and resistance to chemotherapy was generated. In addition to modifying EZH2 activity by palmitoylating GSK3β, EZH2 activity can also be affected by palmitoylation directly. For example, in p53-mutant GSCs, ZDHHC5 palmitoylated EZH2 at Cys571 and Cys576 [52]. After being palmitoylated, palmitoylation blocked phosphorylation of EZH2 and, thus, its activity in mediating methylation of H3K27me3. As a consequence, loss of H3K27me3 released the suppressed expression of stemness markers cluster of differentiation (CD)133 and SOX2, thereby increasing the neurosphere formation capacity of GSCs. Further studies found that ZDHHC5-mediated EZH2 palmitoylation is regulated by mutant p53 as it transcriptionally induced expression of ZDHHC5 by interacting with nuclear transcription factor NF-Y. More recently, it was observed that P53-induced ZDHHC5 expression can also be enhanced by treatment of propofol, an agent that induces local anesthetics [108]. Consequently, upregulated ZDHHC5 led to increased EZH2 palmitoylation, resulting in reduced H3K27me3 at promoter regions of genes that regulate stem cell potency and increase expression of these genes.

Hyperactive transforming growth factor-beta (TGF-β) signaling has also been viewed as a signature event in mesenchymal glioblastoma [109,110]. Downstream of TGF-β, the TGF-β receptor phosphorylates transcription factor SMAD3, inducing expression of genes including leukemia inhibitory factor (LIF) and platelet-derived growth factor-beta (PDGF-β) [111,112,113]. By doing so, TGF-β induces self-renewal and promotes proliferation of glioma-initiating cells [112,113,114]. Recently, Fan et al. reported that palmitoylation of SMAD3 mediated by ZDHHC19 is essential in transducing TGF-β signaling in GSCs [115]. Specifically, palmitoylation of SMAD3 at Cys421 mediated the translocation from cytosol to the nucleus.

OCT4A, also known as POU5F1, is a key transcription factor in the self-renewal, proliferation, and differentiation of stem cells [116,117]. Palmitoylation of OCT4A mediated by ZDHHC17 is essential for protecting OCT4A from degradation and, thus, maintaining its protein stability [118]. Consequently, palmitoylation of OCT4A retained the stemness of GSCs by binding to the enhancer of SOX2, a key gene for maintaining the tumorigenicity of GSCs; thus, it promoted SOX2 expression. Targeting OCT4A palmitoylation using a competitive inhibitor that contains the OCT4A palmitoylation sequence effectively inhibited palmitoylation of OCT4A and, thereby, reduced the tumorigenesis in vivo. Palmitoylation of OCT4A mediated by ZDHHC17 was identified as a promising therapeutic approach toward effectively eliminating cancer-initiating cells.

The involvement of ZDHHC15 and ZDHHC18 in GSCs was revealed recently [119,120]. Knockdown of ZDHHC15 reduced the capacity of GSCs to form neurospheres, indicating that ZDHHC15 is essential for GSCs’ self-renewal [119]. Immunoprecipitation-coupled with mass spectrometry analysis identified GP130, an IL-6 receptor subunit, as a substrate of ZDHHC15 in GSCs. Further mechanism analysis found that palmitoylation mediates the cell membrane localization of GP130 and, thus, activates IL-6/STAT3 signaling to maintain GSCs’ renewal ability. Whereas, for ZDHHC18, although no palmitoylation targets were identified, high expression of ZDHHC18 was observed in mesenchymal GSCs [120]. Mechanistic investigation found that, by interacting with the E3 ligase of BMI1, ZDHHC18 blocked degradation of BMI1, a gene contributing to the maintenance and renewal of cancer-initiating stem cells [121,122], and, thus, maintained the self-renewal capacity of GSCs [120].

### 2.4. Palmitoylation Regulates Cancer Cell Migration by Modifying Cytoskeleton-Related Proteins and Cell Adhesion Molecules

In addition to regulating cancer cell growth, palmitoylation also contributes to cancer cell migration and invasion by modifying functions of actin cytoskeletal remodeling related proteins and cell adhesion molecules. For example, palmitoylation of RhoU, an atypical Rho GTPase, at Cys256 is essential for migration of prostate cancer cells [123,124]. Mechanistic analysis found that palmitoylated RhoU increased cell migration ability by interacting and, thus, stabilizing Cdc42, a gene that regulates cell spread area [124,125].

Palmitoylation of α-tubulin has also been reported [126,127,128,129]. For example, it was reported that palmitoylation at Cys377 mediates astral microtubule function during nuclear migration in the M phase of the cell cycle. More interestingly, it was revealed that palmitoylation level of α-tubulin can also been regulated. For example, androgen treatment increased α-tubulin palmitoylation to mediate proliferation of prostate cancer cells, although the molecular mechanism was not fully investigated [130].

Cancer cell migration can also be regulated by modifying palmitoylation of cell adhesion molecules. For example, palmitoylation of cell adhesion molecules CD44 and MCAM prevented melanoma cell invasion [131]. On the contrary, depalmitoylation of CD44 and MCAM mediated by APT1 resulted in increased invasion.

### 2.5. Cancer Cells Use Palmitoylation to Overcome Nutrient Deficiency

GLUT1 is a widely expressed glucose transporter responsible for the constant uptake of glucose [132]. Numerous studies have also shown that GLUT1 is essential for cancer growth [133,134,135]. In glioblastoma cells, site-directed mutagenesis showed that GLUT1 is palmitoylated at Cys207 and disruption of palmitoylation abolished localization of GLUT1 on the plasma membrane [136]. Expression of GLUT1-WT but not the palmitoylation defective form of GLUT1, GLUT1-C207S, restored the reduction in glucose uptake induced by knockout of GLUT1. Furthermore, blocking GLUT1 palmitoylation mediated by knockout of ZDHHC9, the enzyme that palmitoylates GLUT1, impaired glycolysis and reduced GBM tumor growth. These observations indicate that palmitoylation is required for GLUT1 to localize on the plasma membrane and, thus, conduct its function in mediating glucose uptake and tumor growth.

Malate dehydrogenase 2 (MDH2) catalyzes the reversible reaction of malate to oxaloacetate in the TCA cycle [137]. Aberrant MDH2 function has been found to be associated with malignancy of cancer [138,139]. Palmitoylation of MDH2 was firstly identified in 2008 from a palmitoylome-proteomic study [140]. Recently, Pei et al. found that palmitoylation of MDH2 at Cys138 increased its binding affinity with its coenzyme NAD+ in order to maintain the function of glycolysis and mitochondrial respiration in ovarian cancer cells [141]. MDH2 interacted with ZDHHC18 and exogenous expression of ZDHHC18 increased palmitoylation levels of MDH2, indicating that ZDHHC18 is the palmitoyl transferase responsible for MDH2 palmitoylation. Of note, palmitoylation of MDH2 mediated by ZDHHC18 was stimulated by glutamine deprivation. These data revealed how cancer cells use palmitoylation and mitochondrial respiration for its growth to adapt to the nutrition-deficient tumor microenvironment.

### 2.6. Palmitoylation Negatively Contributes to Cancer

In addition to positively contributing to cancer progression, the negative function of palmitoylation in cancer growth has also been revealed. Palmitoylation conducts its cancer suppression function by suppressing the function of oncoproteins, maintaining functions of tumor suppressors, or modifying growth-signaling transduction.

#### 2.6.1. Palmitoylation Negatively Regulates Oncoprotein Functions

Astrocyte elevated gene-1 (AEG-1) is an oncogene that is over-expressed in a wide variety of cancers [142,143,144,145]. Palmitoylation of AEG-1 in several physiological contexts was initially reported by Zhou et al. [146] (Figure 6a). From this research, they found that AEG-1 is palmitoylated at Cys75 and a palmitoylation-defective form of AEG-1 enhanced hepatocellular carcinoma progression in vivo. Mechanistically, palmitoylation negatively regulates AEG-1 function by reducing the stability of AEG-1 and by reducing its affinity with its interacting proteins. Further, a different research group identified ZDHHC6 as the main palmitoyl transferase for AEG-1 and found that CRISPR/Cas9 knock-in mice with a palmitoylation-defective form of AEG-1 had increased signaling pathways and regulators that contribute to cell proliferation, motility, angiogenesis, and lipid accumulation [147]. Together, palmitoylation plays a negative regulatory role on AEG-1 function.

By transducing PI3K/AKT and RAS/MAPK signaling, receptor tyrosine kinase FLT3 plays an important role in the development of hematopoietic progenitors [148]. Consequently, mutation of FLT3 is one of the causes of cancer. For example, internal tandem duplication within FLT3 (FLT3-ITD) confers constitutive activation of FLT3, represents one of the most frequent mutations in acute myeloid leukemia (AML), and correlates with a poor prognosis [149]. Palmitoylation of FLT3-ITD mediated by ZDHHC6 was identified in AML, and palmitoylation redirected the localization of FLT3-ITD from the plasma membrane to ER (Figure 6b) [150]. Mutation of Cys563 maintained FLT3-ITD on the plasma membrane to conduct its function in activating downstream signaling. Further, inhibition of depalmitoylation with palmostatin B (palm B), a pan-depalmitoylase inhibitor, not only reduced proliferation of FLT3-ITD+ AML cells but also synergized FLT3-ITD+ AML cells to gilteritinib, a FLT3 kinase inhibitor, treatment. Together, these results suggest that palmitoylation plays a repressive role for FLT3.

#### 2.6.2. Palmitoylation Maintains Functions of Tumor Suppressors

G protein-coupled receptors (GPCRs) are a group of membrane proteins that convert extracellular signals into intracellular responses, including responses to growth signaling; neurotransmitters; as well as responses to vision, olfaction, and taste signals [151]. Due to its critical roles in enhancing DNA repair, mutations of the melanocortin-1 receptor (MC1R), a GPCR, have been correlated to a higher risk of melanoma. Palmitoylation of MC1R at Cys78 and Cys315 was identified in melanocytes, and the C315S mutant, a palmitoylation-defective form of MC1R, promoted melanomagenesis [152] (Figure 6c). Palmitoylation of MC1R is mainly mediated by ZDHHC13, and hyper-palmitoylation of MC1R mediated by ZDHHC13 also prevented melanomagenesis. Further, patients with high ZDHHC13 are correlated with better survival, and inhibition of APT2, the MC1R depalmitoylation enzyme, effectively suppressed melanomagenesis by blocking MC1R palmitoylation [153]. Together, these results highlighted a central role of MC1R palmitoylation in protection against melanoma.

TP53 is one of the most extensively studied tumor suppressor genes whose multifaceted mechanisms involve apoptosis, DNA repair, genomic stabilization, and angiogenesis [154]. A recent study found that TP53 is the key mediator for ZDHHC1-induced breast cancer suppression (Figure 6d). Five palmitoylation sites—Cys135, Cys176, Cys182, Cys275, and Cys277—were identified in TP53. Mutation of Cys135, Cys176, and Cys275 significantly blocked nuclear translocation of TP53 and the expression of TP53 downstream targets, e.g., P21 and BAX, resulting in the promotion of tumor growth. Together, by modifying TP53 nuclear translocation, palmitoylation conducts its tumor suppression function [155].

SET domain-containing 2 (SETD2) is a histone lysine methyltransferase. By mediating the methylation of H3K36me3, SETD2 contributes to DNA damage response (DDR) by recruiting RAD51 to DNA double strand break sites. As a functional DDR is critical for maintaining genome integrity and preventing tumor development [156], SETD2 is considered as a putative tumor suppressor gene in cancers [157,158]. In EGFR-amplified glioblastoma, palmitoylation of SETD2 mediated by ZDHHC16 protects SETD2 from degradation and, thereby, facilitates its role in mediating DNA damage response and repressing cancer initiation (Figure 6e) [159].

GNA13 encodes one of the alpha subunits of the heterotrimeric G proteins that transduce signals of GPCR. By negatively regulating the expression of BCL2, GNA13 has also been identified as a tumor suppressor in B-cell lymphoma [160]. Palmitoylation of GNA13 was initially reported in early 2000 (Figure 6f). The study revealed that GNA13 is palmitoylated at Cys14 and Cys18. The wild type but not the palmitoylation-defective form of GNA13 localizes at the plasma membrane and, thus, transduces Rho-dependent signaling [161]. Moreover, expression of GNA13WT but not the palmitoylation-defective form of GNA13 inhibited proliferation of B-cell lymphoma, suggesting that palmitoylation of GNA13 is required for its tumor suppression function [160].

#### 2.6.3. Palmitoylation Suppresses Cancer by Reducing Growth Signaling

In addition to repressing tumors by directly affecting functions of cancer-related protein, palmitoylation can also block tumor growth by modifying growth signaling. For example, palmitoylation at Cys44, Cys45, and Cys47 of the small CTD phosphatase 1 (SCP1) leads to its translocation from the nuclear to plasma membranes [162]. As a result, membrane-located SCP1 dephosphorylates AKT at serine 473, leading to suppressed angiogenesis and decreased tumor growth of lung carcinoma in xenograft mice.

## 3. Targeting Protein Palmitoylation for Cancer Treatment

Palmitoylation has been proposed to be targeted from three directions: targeting ZDHHC enzymes, blocking substrate palmitoylation, and preventing depalmitoylation, although challenges exist.

Targeting ZDHHCs for disease treatment has attracted increased attention. However, as introduced above, many proteins can be palmitoylated by more than one ZDHHC enzyme. Therefore, inhibiting a single ZDHHC cannot fully block the palmitoylation of the substrates. One potential solution to this problem is to search for pan-ZDHHC inhibitors that can block the function of several enzymes required for the palmitoylation of a single target. Also, a complementary ZDHHC-agnostic approach to eliminate the problem of functional redundancy between individual ZDHHC enzymes is to identify compounds that directly prevent substrate palmitoylation via irreversible covalent modification of individual cysteine residues. Further, although ZDHHCs contain a conserved zinc finger (DHHC) domain, few ZDHHC inhibitors are available and no therapeutic drugs that target specific ZDHHCs have been approved to date. The few reported broad-spectrum ZDHHC inhibitors include compound V [163], tunicamycin [164], cerulenin [165], and 2-bromopalmitate (2-BP) [166] (Table 1). Among these inhibitors, 2-BP has been used in pre-clinical studies to validate the concept that ZDHHC protein inhibition can promote cancer death [43,167,168]. However, 2-BP is not selective for individual ZDHHC enzymes, and off-target acylation of other intracellular proteins increases the risk of unknown side effects and limits its potential as a therapeutic candidate for disease treatment [169,170,171,172]. In 2021, another broad-spectrum ZDHHC inhibitor, cyano-myracrylamide (CMA), was identified [173]. Although developing treatments to target ZDHHCs has limitations, as described above, researchers continually make new discoveries to overcome the challenges. For example, last year, Qiu et al. identified that artemisinin (ART), a clinically approved antimalarial endoperoxide nature product, can be used as a ZDHHC6 inhibitor; although, it is unclear if ART also inhibits the function of other ZDHHCs [174].

Rather than inhibiting protein palmitoylation, in some cases, it may be more effective to prevent depalmitoylation, although each acyl-protein thioesterase can depalmitoylate more than one protein (Table 1). For example, inhibition of APT enzymes suppressed tumor formation by promoting the proper localization of SCRIB and enhancing activity of MC1R [152,186,187]. APTs, in general, are highly druggable targets, and a handful of inhibitors have been developed to target APTs. Among those, palmostatin B (Palm B) [176] and hexadecylfluorophosphonate (HDFP) [10] are being widely used in research as they are broad-spectrum serine hydrolase inhibitors that target all depalmitoylases. Although significant overlapping in substrates is observed in different APTs, selectivity in substrates among different APTs has also been revealed [31,32,33]. Furthermore, inhibitors targeting different APTs in the selective way are also well known. For example, in 2013, the first selective inhibitors for APT1 and APT2, ML348 and ML349, were identified [9,188]. More recently, several inhibitors have been revealed to inhibit specific depalmitoylase selectively. For instance, in oral squamous cell carcinomas (OSCC), the expression level of PPT1 can be reduced by erianin, a natural bibenzyl compound, although the molecular mechanism was not fully characterized [189]. GNS561, a specific PPT1 inhibitor, by itself, is able to effectively inhibit the progression of liver cancer [177,190]. Furthermore, another selective and potent PPT1 inhibitor, DC661, not only impaired tumor growth [180] but also enhanced the response of liver cancer cells to treatment of sorafenib, a multi-kinase inhibitor [191], and enhanced the antitumor activity of the anti-PD-1 antibody in melanoma [192]. Dimeric quinacrines 661 (DQ661) was also identified as an inhibitor for PPT1 and, thus, can be used to inhibit cancer growth [181]. More recently, from a serine-hydrolase-directed compound library screening, ABD957 was identified to have selective inhibition on ABHD17 [179].

To target palmitoylated proteins specifically, recently, a few studies have developed customized peptides to compete with the palmitoylation of specific proteins and, thus, block the palmitoylation of specific proteins. For example, Chen et al. showed a cell-penetrating peptide that contains the palmitoylated sequence of OCT4A—a key transcription factor in the self-renewal, proliferation, and differentiation of stem cells—could function as a competitive inhibitor to effectively inhibit the palmitoylation of OCT4A [118]. Similarly, Yao et al. developed a competitive inhibitor of PD-L1 palmitoylation by fusing green fluorescent protein (GFP) to the palmitoylation motif of PD-L1 [90]. Using the same concept, a peptide that fuses GFP with the palmitoylation motif of PCSK9, a gene that plays a critical role in anti-tumor immune responses, also showed inhibition on palmitoylation of PCSK9 both in vitro and in vivo [66].

Different from the competitive inhibitors, inhibitors targeting the palmitoylation pocket have also been reported. For example, several inhibitors have been developed to target the transcriptional enhanced associate domain (TEAD) palmitoylation pocket [185,193,194,195] (Table 1). Serving as the receptor for the downstream effectors of the Hippo pathway, YAP and TAZ, TEAD upregulates the expression of multiple genes involved in organ size control and tumorigenesis. Palmitoylation of TEAD at conserved cysteine residues was shown to be required for binding of TEAD to YAP and TAZ in a variety of cancer cells [196,197]. Structural analysis found that TEAD palmitoylation is critical for protein folding and stability as the lipid tail extends into a conserved hydrophobic core of the protein [198]. Targeting the palmitoylation pocket of TEAD, several inhibitors have been identified. For example, compounds TM2 and MGH-CP1 were identified as inhibitors that bind to the palmitate-binding pocket (PBP) to suppress TEAD palmitoylation and, thus, function [183,184,185]. Further, from a structure-based virtual ligand screening, JM17 was identified as an inhibitor for TEAD palmitoylation. Treatment of JM17 reduced the stability of TEAD, leading to impaired proliferation; colony formation; and migration of mesothelioma (NCI-H226), breast, and ovarian cancer cells [182].

## 4. Future Perspectives

Although we have learned significantly about palmitoylation, many critical issues remain. For example, although we have showed that the functions of numerous proteins are regulated by palmitoylation, we have little understanding on how palmitoylation and depalmitoylation are regulated. Recently, several studies have suggested that palmitoylation can be regulated by altering the expression level, protein stability, or distribution of enzymes involved in palmitoylation [131,159,199,200,201,202]. Other studies have suggested that palmitoylation can be regulated by extracellular stimulators [130,203,204]. Studying the regulation of palmitoylation can help us understand the roles of palmitoylation in leading to pathological diseases.

Although ZDHHCs display different preferences for different substrate proteins [17,18,205,206], no consensus on palmitoylation sequence motifs has been reached and the mechanism of enzyme–substrate pairs has not been established [207]. Palmitoylated proteins are often substrates for more than one ZDHHC enzyme, while one particular ZDHHC enzyme often has a stronger effect than others on substrate palmitoylation in the cell [19,20,207]. ZDHHCs vary substantially in their palmitoylation activities and their affinities in binding to their cognate substrate proteins [205]. Although ZDHHCs have a similar structure, a few ZDHHCs do have some unique signatures. For example, ZDHHC5 and ZDHHC8 have a long and highly disordered C-terminal tail [208,209], whereas ZDHHC13 and ZDHHC17 possess an ankyrin-repeat domain [205,210]. In the past few years, a new technology leveraging high-density CRISPR screens has been used to identify novel functional domains on a protein [211,212,213]. We expect that this technology can possibly be used to study protein structures of ZDHHCs for palmitoylation research fields.

In terms of relevant diseases, it would be interesting to test if cells under certain pathological conditions have a different list of palmitoylated proteins compared with cells under physiological conditions and, if so, if the palmitoylation state of individual cysteine residues within a given protein vary over time when cells undergo malignant changes. In cancers, 79 out of the 299 cancer drivers have been identified to be palmitoylated [8,42]. For example, palmitoylation of GNAQ/11, a G protein α subunit q/11 polypeptide, was reported in uveal melanoma (UM) that contains a GNAQ/11 mutation in over 85% of patients [214]. Specifically, by controlling the location of GNAQ/11, palmitoylation is essential for GNAQ/11 in mediating growth signaling and, thus, malignant progression of UM. As the research moves forward, we speculate that there will be more studies focused on understanding if palmitoylation affects functions of the critical cancer drivers. For example, similar to UM, over 85% of patients with Ewing sarcoma, a type of bone and soft tissue cancer, contain fusion protein EWSR1-FLI1. It would be of interest to test if palmitoylation also affects the function of EWSR1-FLI1 and if palmitoylation is one of the contributing factors in EWSR1-FLI1-mediated Ewing sarcoma.

Although the lack of antibodies recognizing palmitoylated proteins is hindering our understanding of the implication of a dynamic lipid modification of proteins in cell signaling and regulation, the development of chemical approaches to study protein palmitoylation has revolutionized the understanding of the field [34,36,215]. We foresee that identification of palmitoylated proteins at the proteome level [216] will help the field in understanding the roles of palmitoylation in leading to pathological consequences at the systematic level. Further, genetic screens, e.g., CRISPR screens, have been used to identify vulnerable targets for cancer treatment [217]; applying genetic screens in palmitoylation research would help us test if enzymes involved in palmitoylation can be vulnerable cancer treatment targets in various contexts.

## Figures and Tables

**Figure 1 cells-12-02209-f001:**
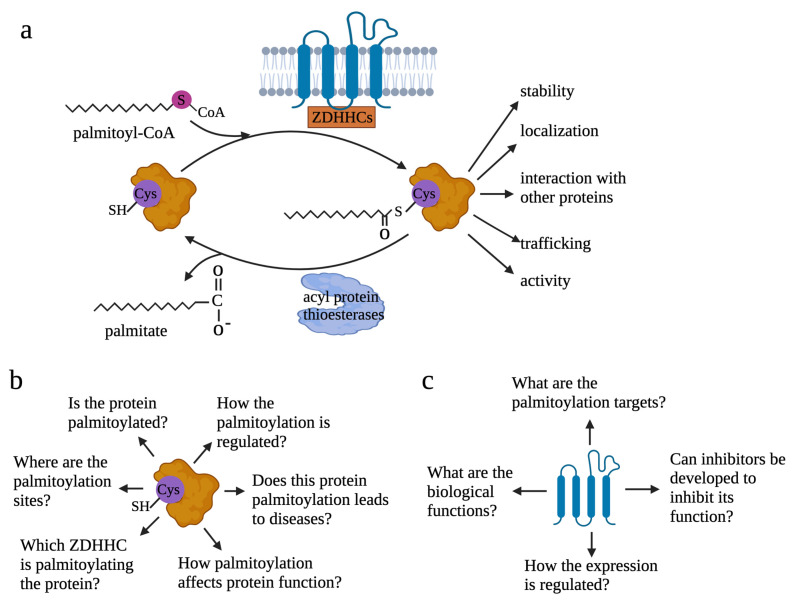
Reversible palmitoylation and questions to be addressed to understand palmitoylation. (**a**) Reversible process of protein palmitoylation. Palmitoyl transferase ZDHHCs transfers palmitoyl-CoA to the cysteine residue of proteins. By adding the lipid tail to proteins, palmitoylation increases the hydrophobicity of proteins and affects protein functions. Depalmitoylation is mediated by acyl-protein thioesterases. (**b**) When studies are focused on targeted proteins, six questions need to be addressed to understand how palmitoylation modifies protein functions: 1—if a specific protein is palmitoylated; 2—where the palmitoylation sites are or which cysteine residues are palmitoylated; 3—which ZDHHC is the main palmitoyl transferase responsible for the palmitoylation of the specific protein; 4—how palmitoylation affects functions of the specific protein; 5—if palmitoylation leads to pathological conditions or contributes to physiological conditions by modulating protein function; 6—how the palmitoylation is regulated. (**c**) When investigations are focused on palmitoyl transferase, four questions need to be addressed: 1—what the biological function of the ZDHHCs in the physiological or specific pathological contexts; 2—for a particular ZDHHC, what the substrates are; 3—if inhibitors can be developed to target the ZDHHCs; 4—how expression of ZDHHCs and how ZDHHCs-mediated palmitoylation are regulated.

**Figure 2 cells-12-02209-f002:**
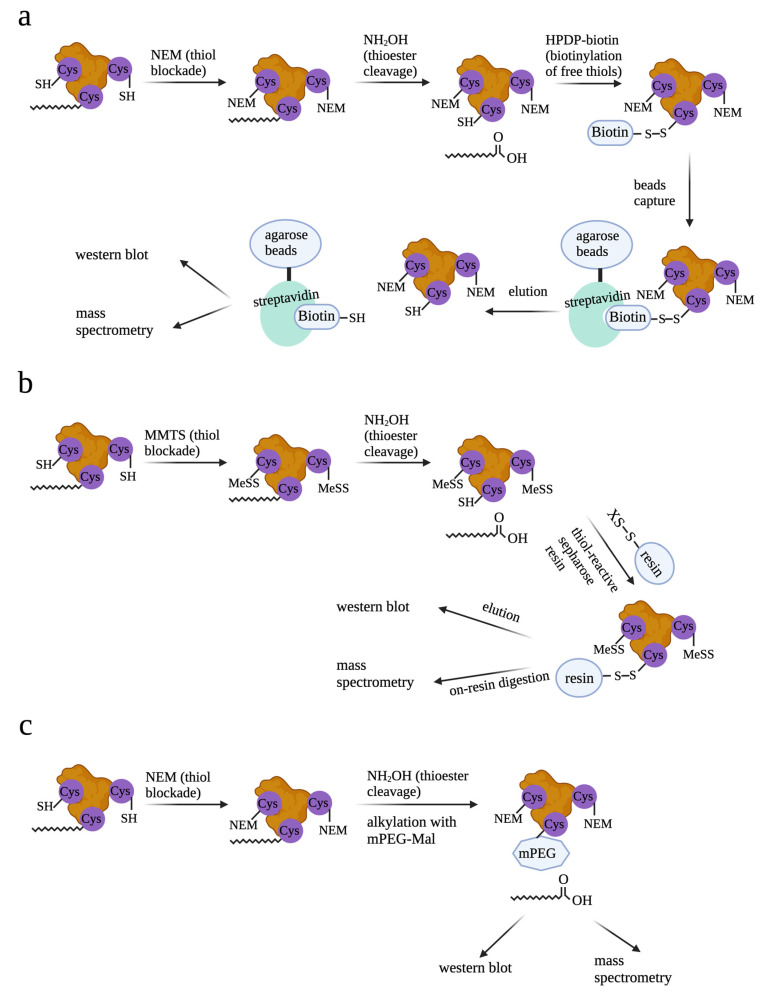
Methods to detect palmitoylated proteins. (**a**) For ABE, free cysteines are blocked with NEM. Thioesters are then cleaved with NH_2_OH and newly generated cysteines are reacted with HPDP-Biotin. Following streptavidin bead enrichment, selectively captured proteins are eluted with reducing agents and then analyzed by Western blot or mass spectrometry. (**b**) Different from ABE, which uses NEM to block free cysteines, acyl-RAC exchange uses methyl methanethiosulfonate (MMTS) to block free cysteines. Further, acyl-RAC uses HPDP-Biotin; newly generated cysteines are reacted with thiol-sepharose resin but not HPDP-Biotin to label the newly generated cysteines after NH_2_OH cleavage. Taking advantage of the on-resin digestion availability, peptides with newly generated cysteines can be captured after elution for mass spectrometry. This allows identification of the specific sites that are palmitoylated. (**c**) With APE, after NH_2_OH cleavage, newly generated cysteines are reacted with 5 kDa methoxy-PEG-maleimide (mPEG-Mal). Due to the big size of mPEG-Mal, proteins which are labeled with mPEG-Mal migrate slower on the Western blot. Proteins with different numbers of mPEG-Mal can also be separated on the Western blot. This allows the identification of numbers of palmitoylation sites of specific proteins.

**Figure 3 cells-12-02209-f003:**
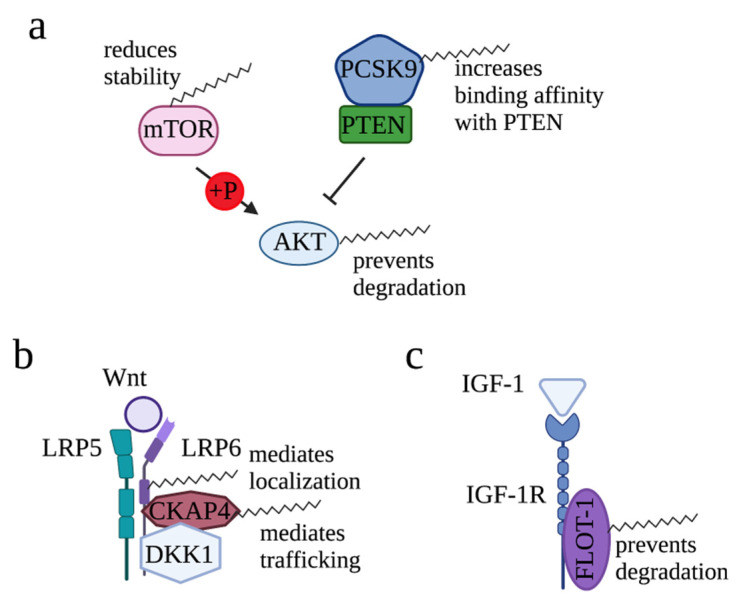
Roles of palmitoylation in modifying AKT, Wnt, and IGF-1 signaling. (**a**) For AKT signaling pathway, palmitoylation reduces mTOR stability, increases binding affinity of PCSK9 with PTEN, and prevents degradation of AKT. (**b**) For Wnt-mediated signaling, palmitoylation maintains plasma membrane localization of LRP6 and mediates CKAP4 translocation to the plasma membrane. (**c**) In IGF-1/IGF-1R signaling, palmitoylation mediates plasma membrane association of FLOT-1, an interacting protein of IGF-1R.

**Figure 4 cells-12-02209-f004:**
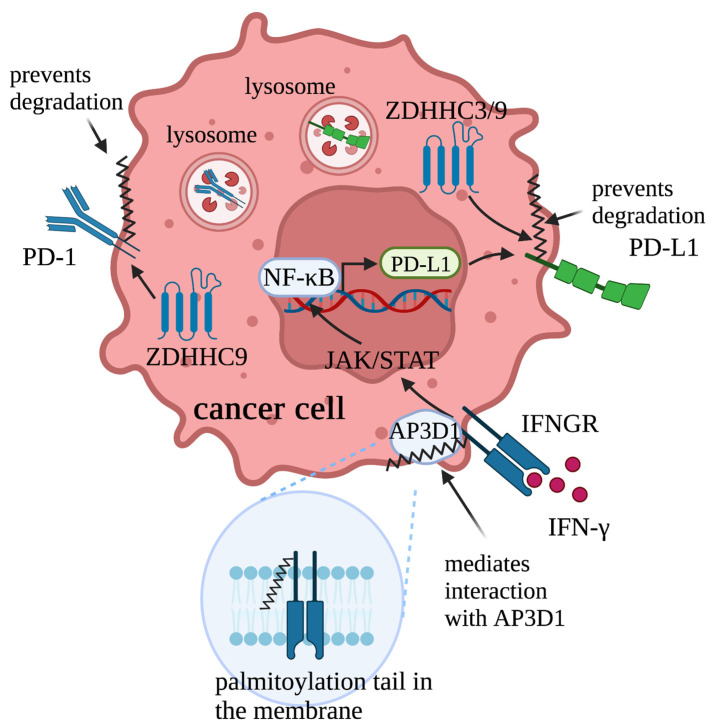
Roles of palmitoylation in regulating IFNγ/IFNGR1-mediated PD-L1/PD-1 signaling against cancer. Palmitoylation regulates this signaling in several layers. Specifically, palmitoylation mediates interaction between IFNGR1 and its binding partner AP3D1, promotes plasma membrane localization of PD-L1, and prevents degradation of PD-1. By doing so, palmitoylation positively contributes to the IFNγ/IFNGR1-mediated PD-L1/PD-1 signaling.

**Figure 5 cells-12-02209-f005:**
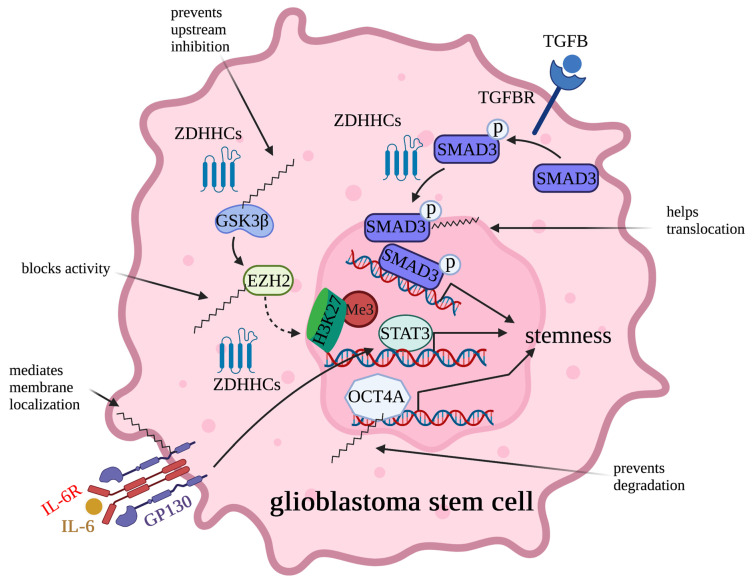
Regulation of the multipotency and proliferation of glioblastoma stem cells (GSCs) by palmitoylation. Palmitoylation regulates functions of GSK3β/EZH2, TGFBR/SMAD3, IL-6/GP130, and Oct4A and, thereby, plays critical roles in maintaining stem cell properties. In detail, palmitoylation increases GSK3β activity by blocking interaction of GSK3β and its upstream inhibitory kinases AKT and S6K, blocks EZH2 in methylating H3K27me3, mediates nuclei translocation of SMAD3, maintains plasma membrane location of GP130, and prevents degradation of OCT4A.

**Figure 6 cells-12-02209-f006:**
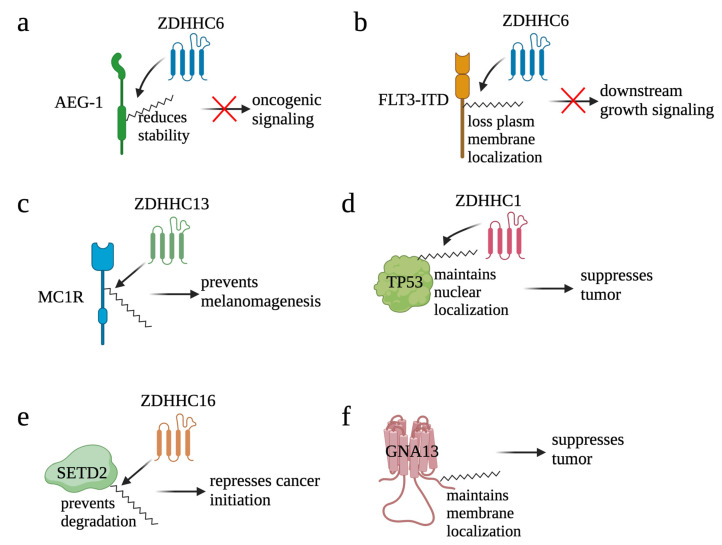
Palmitoylation negatively contributes to cancer. By reducing stability of AEG-1 (**a**) and translocating FLT3-ITD (**b**) from plasma membrane to ER, palmitoylation blocks the oncogenic signaling induced by AEG-1 and FLT3-ITD, respectively. On the other hand, by maintaining function of tumor suppressors, palmitoylation suppresses cancer. For example, palmitoylation helps MC1R in preventing melanomagenesis (**c**), maintaining the nuclear localization of TP53 (**d**), preventing degradation of SETD2 (**e**), and maintaining membrane localization of GNA13 (**f**).

**Table 1 cells-12-02209-t001:** Reported inhibitors to target ZDHHCs and acyl-protein thioesterases.

Categories	Targets	Inhibitors	Structures	References
ZDHHCs	All ZDHHCs	2-Bromopalmitic acid (2-BP)	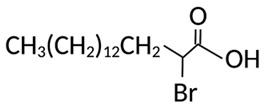	[166,170,171,175]
All ZDHHCs	N-cyanomethyl-N-myracrylamide (CMA)	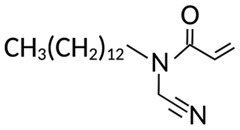	[173]
ZDHHC2, ZDHHC9	Compound V	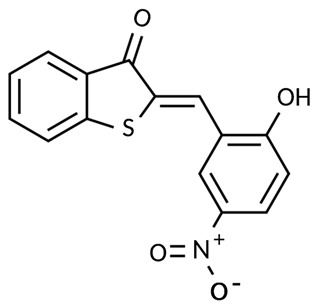	[163]
ZDHHCs	Tunicamycin	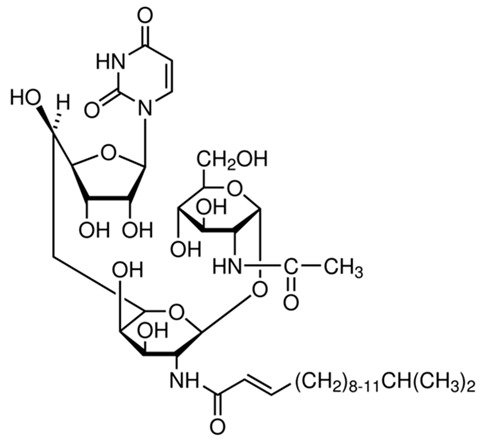	[164]
ZDHHCs	Cerulenin	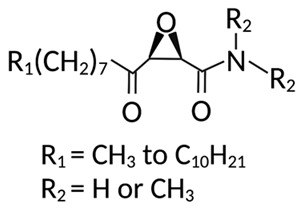	[165]
ZDHHC6	Artemisinin	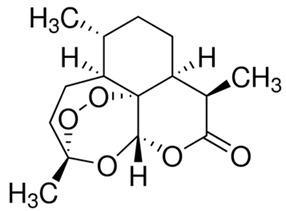	[174]
Acyl-protein thioesterases	APT1	Palmostatin B	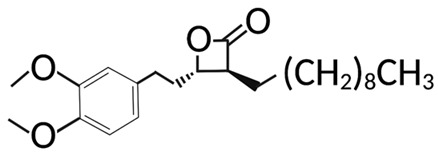	[176]
PPT1	GNS561	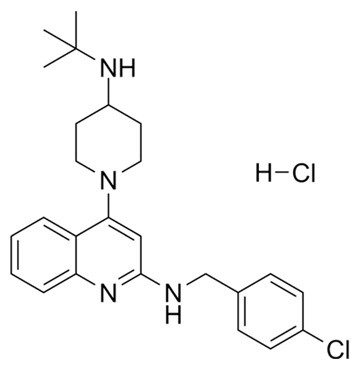	[177]
APT1	ML348	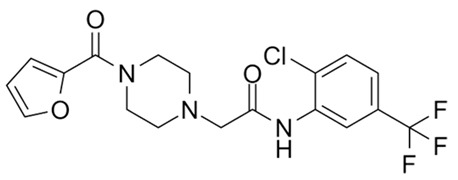	[178]
APT2	ML349	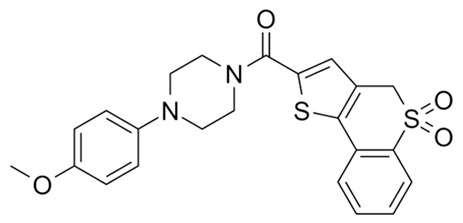	[178]
ABHD17A/B/C	ABD957	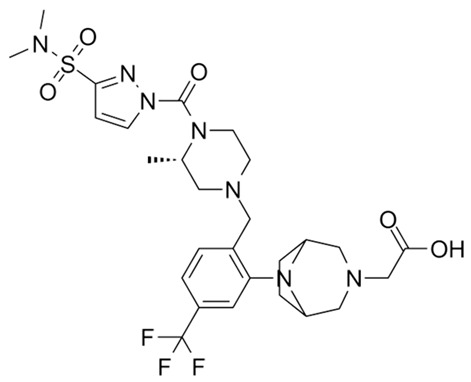	[179]
PPT1	Dimeric chloroquine 661 (DC661)	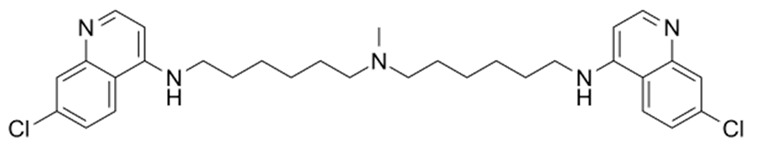	[180]
PPT1	Dimeric quinacrines 661 (DQ661)	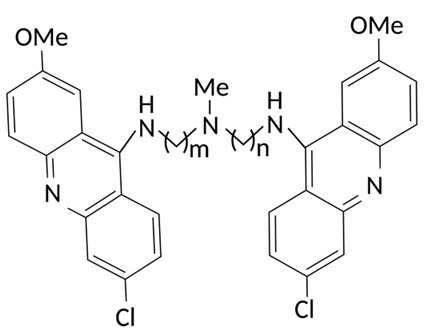	[181]
Palmitoylated protein	TEAD	JM17	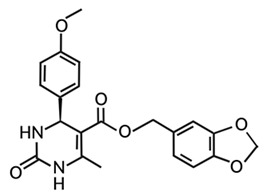	[182]
MGH-CP1	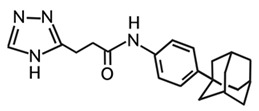	[183]
TM2	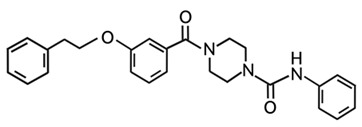	[184]
Compound **2**	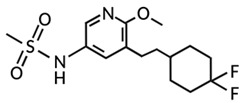	[185]

## Data Availability

Not applicable.

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
