# Peer review of "Diverse Roles of Protein Palmitoylation in Cancer Progression, Immunity, Stemness, and Beyond"

_cells, 2023, doi:10.3390/cells12182209_

Round 1

Reviewer 1 Report

Thank you for putting this review together.  I found it very interesting and I am sure it will be useful to those interested in palmitoylation in cancer. 

There are a few minor corrections that could be done.  

Abstract - line 8 "refers the" should be "refers to the"

Introduction - line 29 refers the" should be "refers to the"

Introduction - line 35 "enables it impacts" should be "enables" or "impacts"

Introduction - line 44 "catalysis activity" should be "catalytic activity"

Introduction - line 49 "comprise of four" should be "are comprised of four"

Introduction - line 50 - remove extra space between (TM) and domains.

Introduction - line 72 "acly-protein" should be "acyl-protein"

Section 2.4 - line 413 "provided cell migration property by" should maybe be "increased cell migration"?  I am not sure what is meant by property.

Section 2.6.1 - line 455 That is not a sentence.  Perhaps "Astrocyte elevated gene-1 (AEG-1) is an oncogene..."

Table 1. Inhibitor column - 2-Bromoplmitic acid (2-BP) should be 2-Bromopalmitic acid (2-BP)".

Page 16 - line 561 - "palmostain B" should be "palmostatin B"

Reviewer 2 Report

Post translational medication of protein plays important role in cell physiology. Different types of protein medication plays different role in maintaining the cell physiology. Here in this review the author has shown how pamitoylation of different proteins plays role in cell physiology and very extensive information has given on palmitoylation and its role in cancer. The author has also mentioned how palmitoylation has both negative and positive effect of cancer succession. The author has explained how a protein gets palmotylated and how its effect AKT and Wnt and few other important pathway in cancer progression. Here the author also mentioned how targeting a palmitoylated protein can help in cancer treatment. The author has elaborated the current studies in palmitoylation and its scope in the field of research specially when cancer in concerned. Overall the review is very informative and will help all interested in protein post translation medication.

Reviewer 3 Report

In this review article, Li et al. describe recent studies linking protein palmitoylation with cancer progression. After initially defining and generally summarizing protein palmitoylation, the authors delve into detail about diverse signaling pathways and the connection between cancer progression or treatment and protein palmitoylation. The breakdown of the subsections based on “growth signaling”, “cancer immunology”, and “cancer STEM cells” followed by descriptions of targeting palmitoylation for cancer targeting and treatment, give the article a clear structure. The figures accompanying the article are also strong and provide a clear graphical representation of the written descriptions. A couple of  figures need minor revisions and a few descriptions need expansion, see detailed comments below, but with these minor revisions, this review should be ready for publication in Cells.

1.     Figures and tables

a.     Figure 1A: The palmitoylation marks in part A both have the wrong number of carbon atoms. The free palmitate has 17 carbons and an awkward chemical formatting to the carboxylic acid. The thioester linkages have 15 carbons. Please revise to make them all 16 carbons.

b.     Figure 1B and 1C: The statements should be formatted into questions as stated in the written text and as labeled in the figure legend. The current formatting with some questions and some broken statements is awkward.

c.     Figure 2: There are only 15 carbons on these palmitoylation marks.

d.     Figure 5?: Figures 3 and 4 are a great supplement to the written text. A Figure 5, similar to the cartoon figures in Figures 3 and 4, should be added to support the written description in section 2.6 about how palmitoylation negatively contributes to cancer.

e.     Table 1: The structures of the TEAD inhibitors (lines 587 – 602) should be added into Table 1.

2.     The sections on depalmitoylases are too short and contain misleading statements.

a.     Lines 71-75: The introduction section on depalmitoylases is too short and contains generalizations that are incorrect.

                                               i.     The authors have lumped together multiple different classes of depalmitoylases with distinct cellular locations, cellular substrates, and biological functions without any differential descriptions. The last sentence in this section is especially misleading, as the motifs that they recognize are known for some of these depalmitoylases like APT1 and APT2 (see Kathayat, Rahul S., Pablo D. Elvira, and Bryan C. Dickinson. "A fluorescent probe for cysteine depalmitoylation reveals dynamic APT signaling." Nature chemical biology 13.2 (2017): 150-152; Kathayat, Rahul S., et al. "Active and dynamic mitochondrial S-depalmitoylation revealed by targeted fluorescent probes." Nature communications 9.1 (2018): 334; and Amara, Neri, et al. "Synthetic fluorogenic peptides reveal dynamic substrate specificity of depalmitoylases." Cell chemical biology 26.1 (2019): 35-47.) but the majority of these depalmitoylases were only recently discovered and do not have sufficient characterization to know if they recognize distinct substrates or substrate motifs. This section needs to be expanded and more detail added to provide a richer view of the role of depalmitoylases in this cycle.

b.     Line 556-575. The review returns to depalmitoylases again within the context of novel inhibitors. Similar to the last section, all of these depalmitoylases are lumped together without differentiation of their cellular localization or function. Without this information, there is no connection between these depalmitoylase inhibitors and cancer treatment. The authors need to add more information about the different locations, substrates, and known functions of these different depalmitoylases or they should remove depalmitoylases and their inhibitors from the article.

3.     The article needs to be carefully edited for proper word choice and sentence structure. A few instances are highlighted in comments on the attached pdf, but a more careful final edit needs to be performed on the entire article.

The article needs to be carefully edited for proper word choice and sentence structure. A few instances are highlighted in comments on the attached pdf, but a more careful final edit needs to be performed on the entire article.
